# Radiative anti-parity-time plasmonics

Yumeng Yang[1,2,3,4,5], Xinrong Xie[1,2,3,4,5], Yuanzhen Li[1,2,3,4], Zijian Zhang[1,2,3,4], Yiwei Peng[1,2,3,4], Chi Wang[1,2,3,4], Erping Li [1,2,3,4], Ying Li [1,2,3,4], Hongsheng Chen [1,2,3,4] ✉ & Fei Gao [1,2,3,4] ✉

Space and guided electromagnetic waves, as widely known, are two crucial cornerstones in extensive wireless and integrated applications respectively. To harness the two cornerstones, radiative and integrated devices are usually developed in parallel based on the same physical principles. An emerging mechanism, i.e., anti-parity-time (APT) symmetry originated from non-Hermitian quantum mechanics, has led to fruitful phenomena in harnessing guided waves. However, it is still absent in harnessing space waves. Here, we propose a radiative plasmonic APT design to harness space waves, and experimentally demonstrate it with subwavelength designer-plasmonic structures. We observe two exotic phenomena unrealized previously. Rotating polarizations of incident space waves, we realize polarization-controlled APT phase transition. Tuning incidence angles, we observe multi-stage APT phase transition in higher-order APT systems, constructed by using the scalability of leaky-wave couplings. Our scheme shows promise in demonstrating novel APT physics, and constructing APT-symmetry-empowered radiative devices.

Electromagnetic (EM) waves of spatial and guided forms, as widely known, are two crucial intermediaries in extensive wireless and integrated applications respectively. Therefore, various manipulation technologies are usually developed for the two types of waves in parallel. For example, transformation optics has spawned surface-wave[1–3] and space-wave cloaks[4]. Negative refractions have led to superlens for space waves[5–7] and subwavelength focusing for guided waves[8]. Topology[9] roots in both bound states in the radiative continuums[10] and robust waveguides[11,12].

Decades ago, the concept of parity-time (PT) symmetry[13], which originates from non-Hermitian quantum mechanics, has been introduced into photonics, thus igniting intense interests in non-Hermitian photonics. It has resulted in fascinating phenomena i.e., unidirectional transmission for space waves[14], as well as coherent perfect absorber[15,16], loss-induced transparency[17], and high-performance sensors for guided waves[18,19]. Subsequently, anti-parity-time (APT) symmetry has also been proposed and aroused intense interests across multi disciplines from atomics[20–23], photonics[24–30], classical[31] and quantum[32] circuits,

thermology[33] to magnetics[34], etc. Fundamentally different from PT-symmetry systems, APT-symmetry systems exhibit exotic dynamics, i.e., energy-difference conservation[31,35], thus providing fascinating approaches to harnessing EM waves. On guided waves, APT symmetry has so far resulted in fruitful photonic phenomena, e.g. constant refraction[36], mode switching[37], and enhanced Sagnac effect[38,39]. However, the study on harnessing space waves with APT-symmetry systems is still lacking.

Here, we propose a radiative plasmonic APT design scheme to harness space waves (in Fig. 1). The crucial part of such design is the leaky wave, which plays a significant role in two aspects. Its in-plane propagating component provides imaginary coupling channels to enable the APT system, while its out-of-plane radiation nature enables the interaction of APT systems and space waves. We then experimentally demonstrate the design with subwavelength designer surface plasmonic resonators (DSPRs). The DSPRs host low-frequency surface modes which are analogous to the localized surface plasmons at optical frequencies[40–45]. Exploring the degrees of freedom (DoFs) of

[1]Interdisciplinary Center for Quantum Information, State Key Laboratory of Extreme Photonics and Instrumentation, ZJU-Hangzhou Global Scientific and Technological Innovation Center, Zhejiang University, Hangzhou 310027, China. [2]International Joint Innovation Center, The Electromagnetics Academy at Zhejiang University, Zhejiang University, Haining 314400, China. [3]Key Lab. of Advanced Micro/Nano Electronic Devices & Smart Systems of Zhejiang, Jinhua Institute of Zhejiang University, Zhejiang University, Jinhua 321099, China. [4]Shaoxing Institute of Zhejiang University, Zhejiang University, Shaoxing 312000, China. [5]These authors contributed equally: Yumeng Yang, Xinrong Xie. ✉e-mail: hansomchen@zju.edu.cn; gaofeizju@zju.edu.cn

space waves, we observe two exotic phenomena unrealized previously. Switching polarizations of illuminating space waves, we realize polarization-controlled APT phase transition. Tuning incidence angles, we observe multi-stage APT phase transition in higher-order APT systems, constructed by using the scalability of leaky-wave coupling. Our scheme shows promise in demonstrating novel APT physics, and constructing APT-symmetry-empowered radiative devices.

## Results

### Designer-plasmonic realization of radiative APT systems

Figure 2a shows a designer-plasmonic radiative APT system consisting of two DSPRs. A purple dashed circle indicates an individual DSPR, which is a cylindrical three-layer structure. The top layer is a groove-

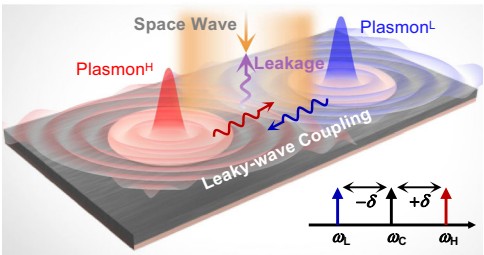

**Fig. 1 | Schematic of the radiative plasmonic APT design.** The system consists of two plasmonic resonators, whose frequencies are oppositely detuned, i.e. up-detuned ($\omega_H$) plasmon[H] and down-detuned ($\omega_L$) plasmon[L]. $\omega_C$ denotes the averaged frequency $\omega_C = (\omega_H + \omega_L)/2$. The two resonators are indirectly coupled by leaky waves, whose out-of-plane radiation leakage enables the interaction of the APT system and space waves.

textured ultrathin copper disk, the middle layer is a dielectric substrate, and the bottom layer is a complete copper plate attached to the substrate. The top surface of the DSPR exhibits a highly-confined dipole mode $\psi_1^{nf}$, whose horizontal decay length is $L_x = 2.9$ mm from the edge of the resonator (in Fig. 2b). The subscripts 1 and 2 denote the left and right DSPR respectively. However, this DSPR mode does not radiate in the $z$-direction, since the upper $x$-polarized dipole and its image generated by the ground plane conceal their radiations in the far-field, due to the deep subwavelength thickness ($h = 2$ mm) of the substrate. Removing the top layer, we obtain background structures, which host surface waves $\psi^p$ propagating along $x$ direction (cross-sectional field in the substrate is shown in Fig. 2c). The dispersion of such surface waves is very close to the light line as shown in Fig. 2d. A slight perturbation (e.g. DSPRs) on the background can transform the surface waves into leaky modes, which not only propagates along $x$ direction (i.e. in-plane radiation), but also radiates in $z$-direction into the free space[46] (i.e. out-of-plane radiation). Consequently, the in-plane component of leaky waves can provide the indirect coupling channels, in addition to the direct coupling through the evanescent field of the two DSPRs. While the out-of-plane radiation leakage enables the interaction of the system and space waves.

In the basis $[\psi_1^{nf}, \psi_2^{nf}]^T$ the coupled designer-plasmonic system is described with a generalized coupled-mode equation:

$$\omega \begin{bmatrix} \psi_1^{nf} \\ \psi_2^{nf} \end{bmatrix} = \begin{bmatrix} \omega_1 + i\gamma_0 & \omega_0(\kappa_{12} + i\chi_{12}) \\ \omega_0(\kappa_{21} + i\chi_{21}) & \omega_2 + i\gamma_0 \end{bmatrix} \begin{bmatrix} \psi_1^{nf} \\ \psi_2^{nf} \end{bmatrix} \quad (1)$$

where $\psi_{1(2)}^{nf}$, $\omega_1$ ($\omega_2$) and $\omega_0$ represent the complex amplitude, the resonant frequency of the DSPR mode and the averaged frequency $\omega_0 = (\omega_1 + \omega_2)/2$, respectively. The coefficient $\kappa_{12}$ ($\kappa_{21}$) quantifies the

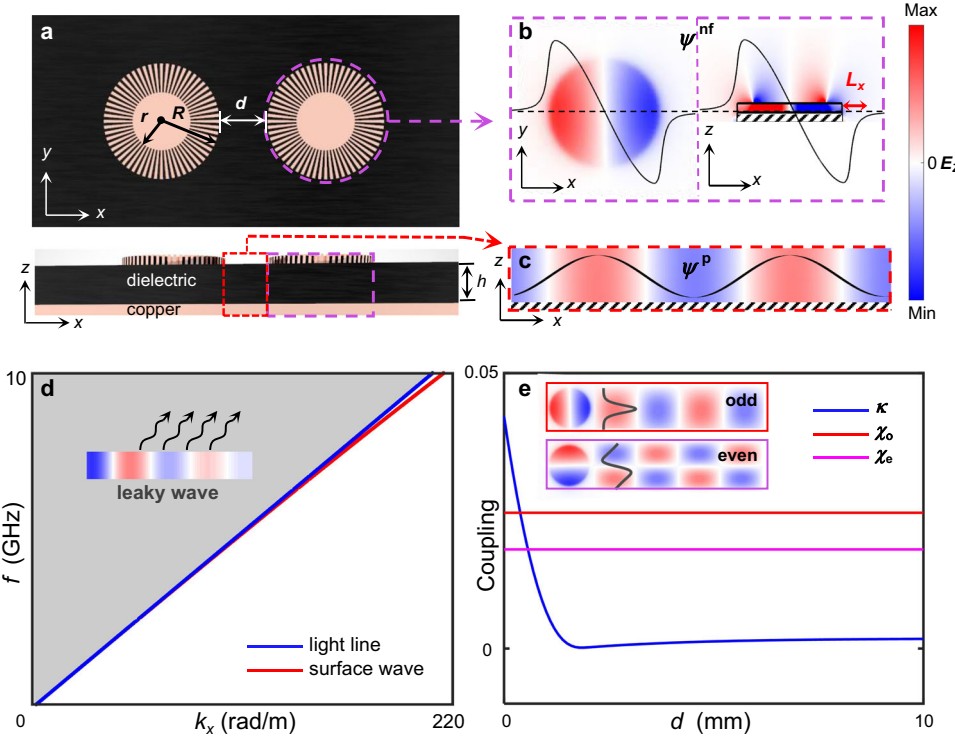

**Fig. 2 | Designer-plasmonic realization of the radiative APT system. a** The top view and the front view of the designer-plasmonic APT system. It consists of two DSPRs with a distance $d$, encircled by the background horizontally. The purple dashed line denotes a single resonator composed of three layers. The top layer is a groove-structured copper plate, standing on a dielectric substrate with the copper coating on its bottom. The background is also constructed with the dielectric substrate coated with copper plating. **b** The $E_z$ component of the simulated DSPR eigenmode ($\psi^{nf}$) at 3.655 GHz on the XY (left) and XZ (right) planes. **c** The $E_z$ component of the surface wave ($\psi^p$) on the XZ plane. **d** The dispersions of the surface wave (red) in the background, and the light line (blue) in the air. The leaky waves lie in the gray region. **e** The distance-dependent strength of calculated direct coupling $\kappa$ (blue) and indirect couplings $\chi_e$ (magenta) and $\chi_o$ (red). The insets depict the mode-parity modulation on leaky waves.

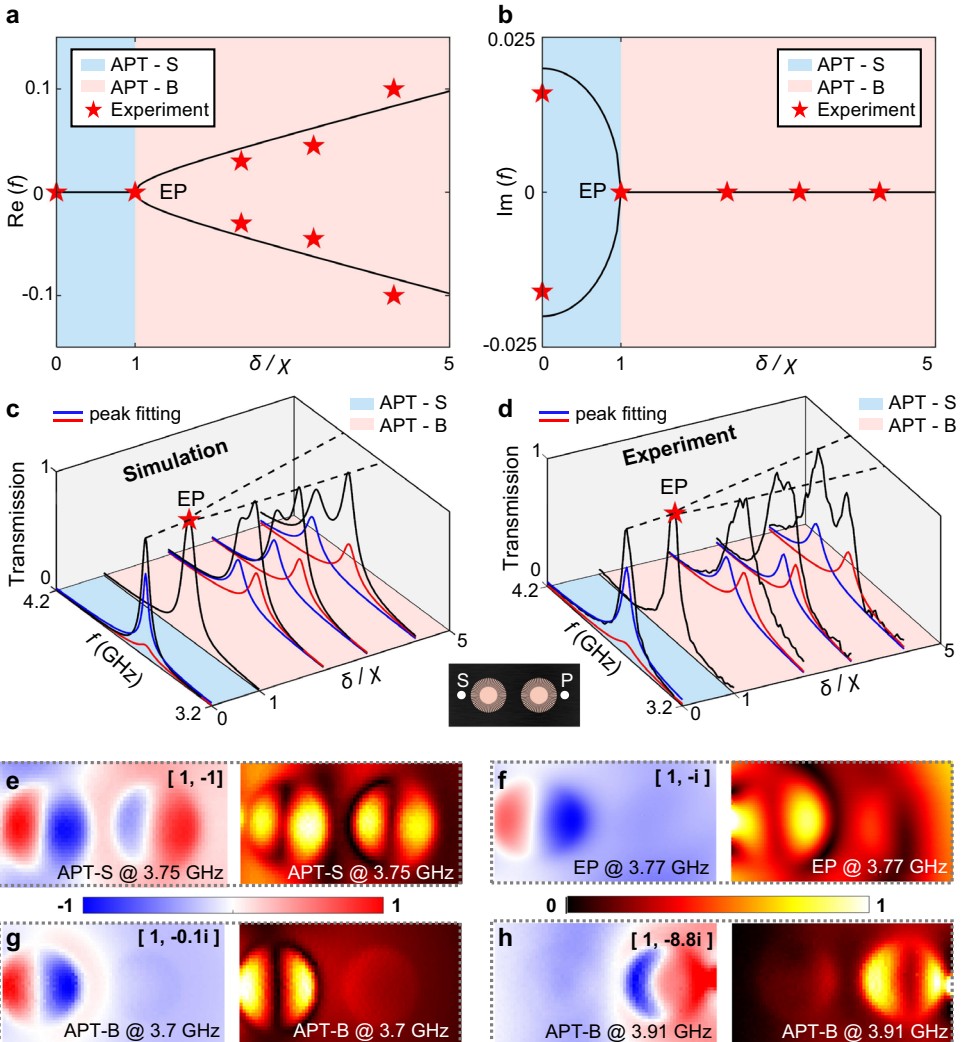

**Fig. 3 | Near-field characterizations of plasmonic APT systems. a, b** The real part (**a**) and imaginary part (**b**) of the calculated eigensolutions of the APT system versus the ratio between the frequency detuning $\delta$ and indirect coupling $\chi$. The blue and red regions denote the APT-S and APT-B phases respectively. The red stars indicate the detected resonance frequencies in experiments. **c, d** The evolution of the transmission spectra (black lines) as $|\delta/\chi|$ varies, obtained with simulations (**c**) and near-field-measured experiments (**d**). Each black line is decomposed into a blue and a red line, according to mode decompositions. The inset shows the setup for both simulations and measurements. The 'S' and 'P' denote the locations of the near-field source and probe respectively. **e–h** The measured near-field distributions of APT-S ($|\delta/\chi| = 0$), EP ($|\delta/\chi| = 1$), and APT-B phases ($|\delta/\chi| = 4.46$), respectively. The distributions of $E_z$ and $|E_z|$ are shown with rainbow and hot colors, respectively.

strength of direct coupling between the two modes, while $i\chi_{12}$ ($i\chi_{21}$) represents their indirect coupling strength through leaky-wave channels[47–49]. Besides the in-plane components of leaky modes which indirectly couple the two DSPRs, the out-of-plane radiation of leaky modes takes the energy of the resonant modes into the ambient space, thus behaving as loss channels (quantified with dissipation rates $\gamma_0$).

Applying energy conservation[50] to the hybrid system including both the plasmonic system and environment, we obtain that coupling coefficients satisfy $\kappa_{12} = \kappa_{21} = \kappa$, $i\chi_{12} = i\chi_{21} = i\chi$ (see Supplementary Information I). Therefore, the system Hamiltonian is expressed as $H = \omega_0 + i\gamma_0 + \omega_0 H_{int}$, where the interaction part is $H_{int} = [-\delta, \kappa+i\chi; \kappa+i\chi, \delta]$. $\delta = |\omega_1 - \omega_2|/(2\omega_0)$ represents the frequency detuning.

We further elucidate the synthesis of APT symmetry in $H_{int}$, which should satisfy the anti-commutation relation, i.e. $\{H_{int}, PT\} = 0$. The parity operator $P$, expressed by Pauli matrix $\sigma_x$, exchanges the spatial positions of the two modes. The time-reversal operator $T$ is given by the complex conjugation. Consequently, such anti-commutation relation requires vanished real couplings, while remaining only the imaginary couplings.

For this purpose, we elucidate these two couplings with the field overlap integral[51]. The indirect coupling through leaky waves is elucidated as $i\chi \propto \int (\psi_{1(2)}^{nf} \cdot \psi_{2(1)}^{p})/(|C_{NF}| \cdot |C_{FF}|) dV$, where normalization constants follow $|C_{NF}|^2 = <\psi_m^{nf} | \psi_m^{nf}>_V$ and $\delta (0)|C_{FF}|^2 = <\psi_m^{p} | \psi_m^{p}>_V$ ($m = 1, 2$). The leaky mode excited by the corresponding resonant mode $\psi_{1(2)}^{nf}$ is approximated with the in-plane propagating wave $\psi_{1(2)}^{p}$. Since the fields of resonant modes are mostly confined in the substrate, the integration is dominated by the parts in the substrate. Since the substrate-field distribution is approximately uniform along the $z$ direction, the volume integral is simplified as $i\chi \propto h \cdot \int (\psi_{1(2)}^{nf} \cdot \psi_{2(1)}^{p})/(|C_{NF}| \cdot |C_{FF}|) dx dy$. It is worth noting that there are two forms of indirect couplings, i.e. even-mode and odd-mode couplings, since the propagating wave $\psi_{1(2)}^{p}$ is modulated by the DSPR mode $\psi_{1(2)}^{nf}$ with different parities (shown in Fig. 2e). Taking an arbitrary line along $x$ direction, the field distributions of $\psi^{nf}$ is approximated as $\cos(\pi y/2R)\sin(\pi x/2R)$ (odd) or $\sin(\pi y/2R)\cos(\pi x/2R)$ (even). Due to the parity modulation by $\psi^{nf}$, $\psi^{p}$ is approximated as $\cos(\pi y/2R)e^{ikx}$ (odd) or $\sin(\pi y/2R)e^{ikx+i\pi/2}$ (even), respectively. $k$ denotes the propagation constant of the in-plane radiation. Therefore, the non-vanished indirect coupling

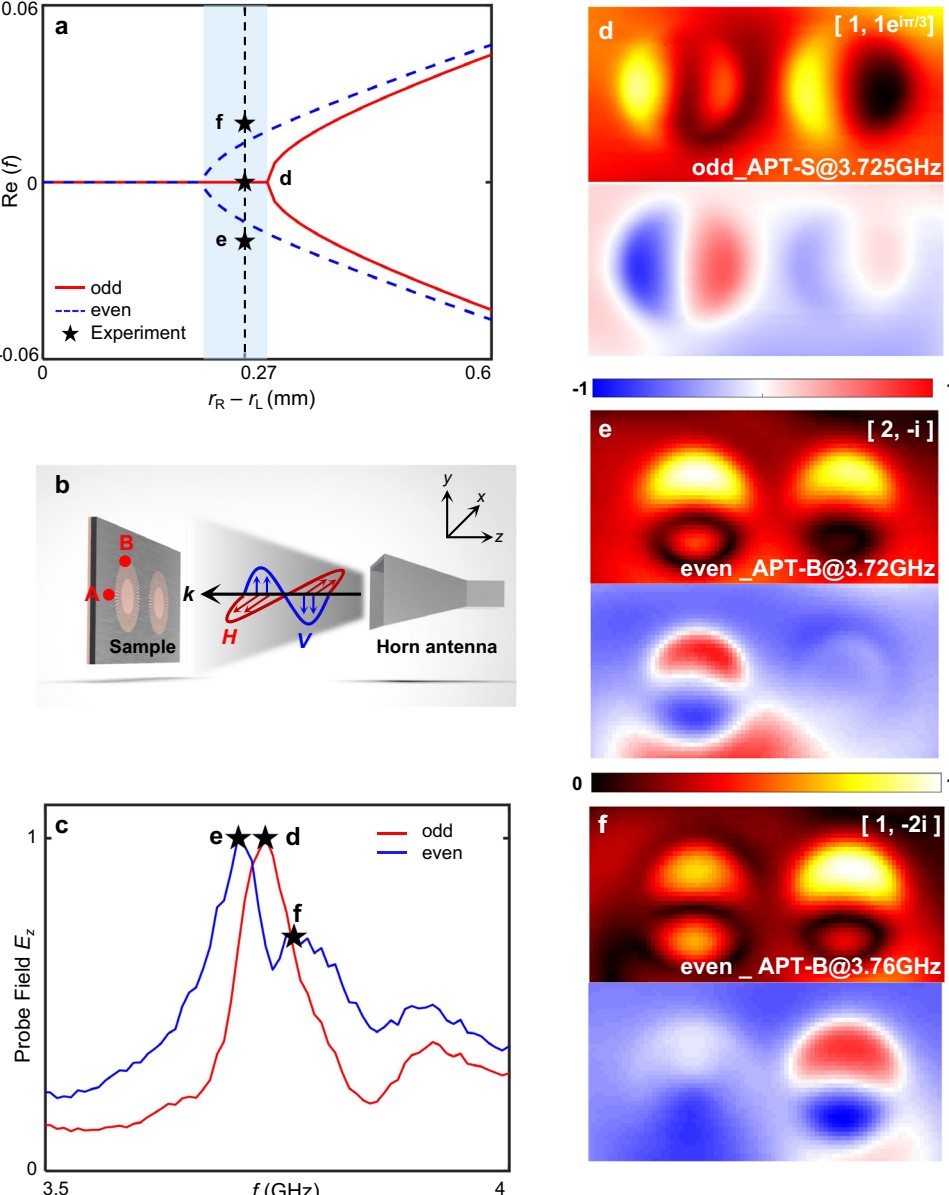

**Fig. 4 | The polarization-controlled APT phase transition. a** The evolution of eigen frequencies as the detuning $r_R - r_L$ changes for odd (red) and even (blue) modes. The stars mark the detected resonance frequencies on the sample $r_R - r_L = 0.27$ mm. **b** The far-field excitation setup. The red and blue arrows denote horizontally (H) and vertically (V) polarized space waves, respectively. Points A and B denote the probe positions for odd and even modes respectively. **c** The experimental transmission spectra under illuminations of H-polarized (red) and V-polarized (blue) plane waves, respectively. The stars mark the resonance-peak positions. **d–f** The field patterns $E_z$ by near-field imaging under H-polarized plane-wave illuminations at 3.725 GHz (**d**), and V-polarized plane-wave illuminations at 3.72 GHz (**e**) and 3.76 GHz (**f**). The distributions of $E_z$ and $|E_z|$ are shown with rainbow and hot colors, respectively.

gives pure imaginary values as shown in Fig. 2e (see Supplementary Information II). In Fig. 2e, the indirect coupling strength is independent of the distance $d$ between two resonators in proper regions (see Supplementary Information II), and $\chi_e$ is less than $\chi_o$. While the direct coupling is also proportional to the field overlap integral $\kappa \propto \int(\psi_1^{nf} \cdot \psi_2^{nf})/(|C_{NF}|^2)dV$ (in Fig. 2e), which decreases as the distance $d$ increases due to the evanescent distance of $\psi^{nf}$. To guarantee vanished direct couplings and retained indirect couplings, we set the edge-edge distance of the two DSPRs sufficiently distant as $d = 10$ mm, which is larger than $2L_x$. Therefore, the interaction Hamiltonian is reduced as $H_{int} = [-\delta, i\chi; i\chi, \delta]$, which satisfies the anti-commutation relation, i.e. $\{H_{int}, PT\} = 0$. It is noteworthy that such photonic indirect coupling, resulting in a plasmonic APT system in the spectral domain, is a mechanism of linear optics, rather than nonlinear optics which usually require high-power input[21,26,29].

The reduced Hamiltonian $H_{int}$ undergoes spontaneous APT-symmetry breaking by sweeping the frequency detuning $\delta$. More specifically, the system changes from APT-symmetric (APT-S) to APT-symmetry-broken (APT-B) phase, when $\delta$ changes from $|\delta/\chi| < 1$ to $|\delta/\chi| > 1$. Three striking features indicate the symmetry-breaking process. Firstly, the eigenvalues $\lambda$ evolve from two pure imaginary values $\pm i(\chi^2 - \delta^2)^{1/2}$ to two pure real values $\pm(\delta^2 - \chi^2)^{1/2}$ (as shown in Fig. 3a, b). Such evolution corresponds to the transition from two degenerate modes with different decay rates (the eigenvalues $\pm i(\chi^2 - \delta^2)^{1/2}$ manifest as high-Q and low-Q mode respectively) to two non-degenerate modes with the same decay rate (the eigenvalues $\pm(\delta^2 - \chi^2)^{1/2}$ correspond to the high-frequency and low-frequency mode respectively). Secondly, the two corresponding eigenmodes of the Hamiltonian change from $[\psi_1^{nf}, \psi_2^{nf}]^T = [1, -(i\delta \mp (\chi^2 - \delta^2)^{1/2})/\chi]^T$ in the APT-S phase to $[1, -i(\delta \pm ((\delta^2 - \chi^2)^{1/2})/\chi)]^T$ in the APT-B phase. Specifically, the APT-S

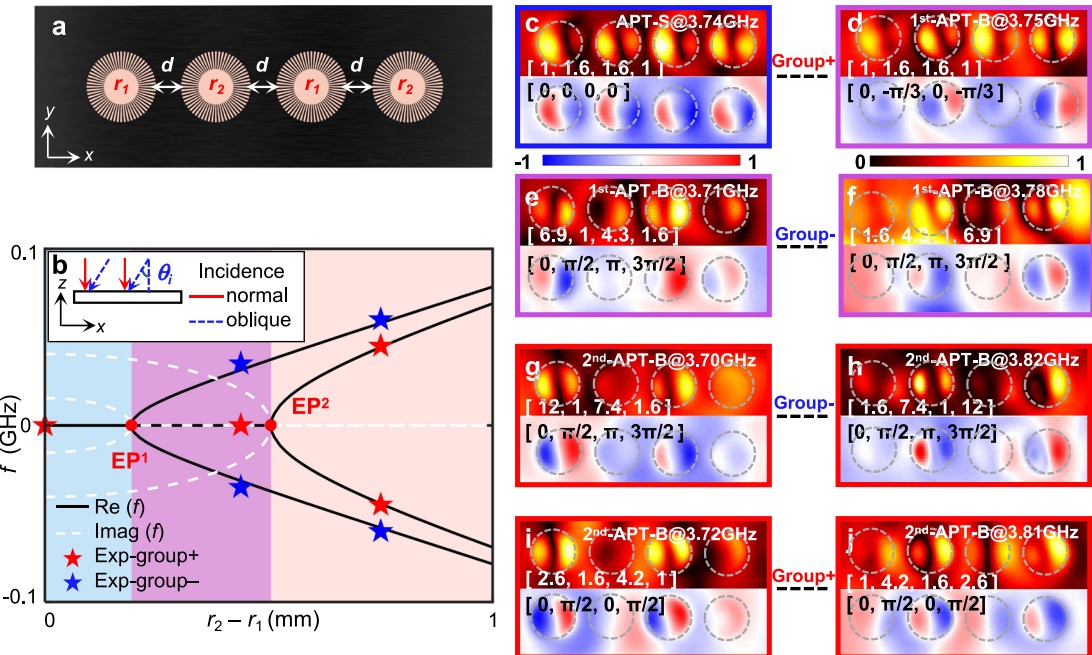

**Fig. 5 | Observation of multi-stage APT phase transitions under space-wave illuminations. a** The top view of the higher-order APT system. From left to right, the inner radii of the four DSPRs are $r_1$, $r_2$, $r_1$, and $r_2$, respectively. **b** The real (black) and imaginary part (white) of the calculated eigenvalues of the system versus the detuning $r_2 - r_1$. The blue, purple and pink regions denote the APT-S, 1st-APT-B and 2nd-APT-B phases, respectively. The red dots represent the EPs. Stars mark the experimentally detected resonance frequencies at $r_2 - r_1 = 0$, 0.4, and 0.7 mm. The inset is the schematic of normal (red) and oblique (blue) illuminations to excite modes in Group+ and Group− respectively. **c–j** The measured field patterns of APT-S at $|\delta/\chi| = 0$ (**c**), 1st-APT-B at $|\delta/\chi| = 1.4$ (**d–f**), and 2nd-APT-B phases at $|\delta/\chi| = 2.4$ (**g–j**), respectively. The distributions of $E_z$ and $|E_z|$ are shown with rainbow and hot colors, respectively.

modes exhibit the unit amplitude ratio ($|\psi_2^{nf}/\psi_1^{nf}| = 1$) and varied phase differences. However, the APT-B modes exhibit non-unit amplitude ratios with a fixed phase difference $\arg(\psi_2^{nf}) - \arg(\psi_1^{nf}) = -\pi/2$ (see Supplementary Information III). Thirdly, the effective Hamiltonian becomes defective at the critical point $|\delta/\chi| = 1$, which is also termed as the exceptional point (EP), and gives a single eigenvalue $\lambda = 0$ and eigenstate $[1, -i]^T$. Therefore, the two-dimensional Hilbert space coalesces into one-dimension space, where the missing dimension is also known as the Jordan vector[52].

## Near-field demonstration of APT phase transition

Without loss of generality, we take the odd mode (in Fig. 2e) to verify the APT phase transition. Five samples with different $\delta$ are realized by changing the inner radius $r_R$ of the right DSPR as 6, 6.3, 6.7, 7.0, and 7.3 mm respectively, while keeping the $r_L = 6$ mm for the left DSPR. Their corresponding frequency detuning parameters $\omega_0\delta$ are extracted as 0, 0.025, 0.061, 0.085, and 0.1115 GHz respectively (see Supplementary Information IV). Under the slight detuning and keeping the edge-edge distance as $d = 10$ mm, the indirect coupling $\omega_0\chi$ is approximately unchanged and extracted as 0.025 GHz from simulation (see Supplementary Information IV for details). Therefore, the sample of ratio $|\delta/\chi| = 0$ corresponds to the APT-S phase, $|\delta/\chi| = 1$ at the EP and $|\delta/\chi| = 2.44$, 3.40, 4.46 in the APT-B phase.

Utilizing near-field measurements (see Supplementary Information V), we further verify the APT phase transition. The near-field transmission spectra are measured by using the point excitation-probe setup (shown in the inset of Fig. 3c). As the ratio $|\delta/\chi|$ decreases, we experimentally observe that two Lorentz peaks gradually merge into one (shown in Fig. 3d), which is consistent with the simulated spectra in Fig. 3c. The peak evolution corresponds to the change from two real eigenvalues to zero and confirms the phase transition from the APT-B to the APT-S phase. Furthermore, the evolution of spectral linewidth also verifies the APT phase transition. The two peaks in the APT-B phase exhibit the same linewidths, which are extracted as 0.036, 0.044, and 0.045 GHz at $|\delta/\chi| = 2.44$, 3.40, and 4.46 respectively. While the spectrum of the APT-S phase ($|\delta/\chi| = 0$) is decomposed into two peaks with different linewidths (0.029 and 0.061 GHz). The linewidth evolution corresponds to the change of the imaginary parts of eigenvalues and further confirms the APT phase transition. The noteworthy phenomenon is that the spectrum measured at the APT-S phase ($|\delta/\chi| = 0$) is dominated by the high-Q mode, while the degenerate low-Q mode is weak. The phenomenon is due to that the high-Q mode is excited with a larger efficiency $(\omega_0\chi + \gamma_0)/2$ under the asymmetric near-field excitation in the existence of background loss, thus dominating the measured near-field spectrum (see Supplementary Information VI).

Using near-field imaging technologies, we further observe the APT phase transition by capturing field patterns at resonance peaks. Comparing the samples of $|\delta/\chi| = 0$ (APT-S) and 4.46 (APT-B), the field patterns exhibit near-unit and non-unit amplitude ratios respectively (in Fig. 3e, g, h), which are consistent with theoretical results. At the EP ($|\delta/\chi| = 1$), the captured field pattern (in Fig. 3f) deviates from the calculated eigenstates $[1, -i]^T$. Such deviation is due to the emergence of the missing eigenstate $[1, i]^T$ in the presence of the background loss $\gamma_0$ (see Supplementary Information VI). These captured field patterns further confirm the APT phase transition. Also, such the detuning-induced APT phase transition is observed under space-wave illuminations (see Supplementary Information VII).

Note that we use different samples here just for the convenience of experimental characterizations. It does not mean this is the only way to achieve the APT phase transition. Since our plasmonic APT system is an open system, the phase transition is also achieved by changing environmental factors, e.g. permittivity (see Supplementary Information VIII and IX). This approach does not require using different samples, and is promising for sensing applications.

## Far-field polarization-controlled APT phase transition

We further investigate the polarization responses of the plasmonic APT system, since the leaky wave links the polarization of space waves to the parity of DSPR modes. Specifically, the horizontal (H) and vertical (V) polarizations are linked to the odd- and even-parity modes, respectively. The even and odd modes exhibit different indirect coupling strengths due to their parity modulations on leaky waves (in Fig. 2e). Therefore, the APT system exhibits two sets of APT phase transitions for odd and even modes (in Fig. 4a). In the blue region of Fig. 4a, the APT system is in the APT-S phase for odd modes while in the APT-B phase for even modes. Since the links between the space-wave polarizations and DSPR-mode parities, we switch the polarizations of illuminating space waves (in Fig. 4b) to induce the APT phase transition. We design the sample ($r_R = 6.27$ mm and $r_L = 6$ mm) in the blue region. A vertical near-field probe is utilized to capture the $E_z$ components of field patterns. We probe the fields at positions A and B (in Fig. 4b) under H- and V- polarized incidence, respectively. When we rotate the space-wave polarization from H to V, the number of resonance peaks changes from one to two (shown in Fig. 4c). The spectral results indicate the polarization-controlled phase transition from APT-S to APT-B, consistent with the theoretical results in Fig. 4a.

The polarization-controlled APT phase transition is further verified by the captured field patterns at the resonance peaks. Under H-polarized incidence, the pattern shows the low-Q mode $[1, e^{i\pi/3}]$ of odd parity at 3.725 GHz, thus confirming the APT-S phase. While under the V-polarized incidence, the captured patterns, corresponding to the even parity, show $[2, -i]$ at 3.72 GHz and $[1, -2i]$ at 3.76 GHz, respectively, thus confirming the APT-B phase.

## Two-stage APT phase transition in higher-order systems

Attributed to the scalability of the leaky-wave coupling, the design mechanism of our radiative APT system is further utilized to study higher-order systems. Moreover, the incoming direction of space waves plays a unique role in identifying high-order APT supermodes. Fig. 5(a) shows the higher-order APT system, which consists of four DSPRs with inner radii $[r_1, r_2, r_1, r_2]$, while other parameters remain the same as the above structures. In the basis $[\psi_1^{nf}, \psi_2^{nf}, \psi_3^{nf}, \psi_4^{nf}]^T$, the Hamiltonian of this system is expressed as $H = \omega_O + i\gamma_O + \omega_O H_4$.

$$H_4 = \begin{bmatrix} -\delta & i\chi & 0 & 0 \\ i\chi & \delta & i\chi & 0 \\ 0 & i\chi & -\delta & i\chi \\ 0 & 0 & i\chi & \delta \end{bmatrix}$$

The fourth-order effective Hamiltonian $H_4$ obeys the four-dimensional anti-commutation relation $\{H_4, P_4 T\} = 0$, where the four-dimensional parity operator is $P_4 = [0, 0, 0, 1; 0, 0, 1, 0; 0, 1, 0, 0; 1, 0, 0, 0]$.

During sweeping the frequency detuning $\delta$, the higher-order APT system undergoes two-stage APT-symmetry breakings (in Fig. 5b), marked by the two distinct EPs, i.e. $|\delta / \chi| = (\sqrt{5} \mp 1)/2$. When $|\delta / \chi| < (\sqrt{5} - 1)/2$, four degenerate modes differ with the imaginary parts of their eigenvalues $\lambda_{1,2,3,4} = \pm i[-\delta^2 + ((3 \pm \sqrt{5})\chi^2/2)]^{1/2}$. We term this region as the APT-S phase, where all four degenerate modes can be classified into two groups according to their fixed amplitude ratios, i.e. $|\psi_1^{nf}| : |\psi_2^{nf}| : |\psi_3^{nf}| : |\psi_4^{nf}| = 1 : (\sqrt{5} + 1)/2 : (\sqrt{5} + 1)/2 : 1$ as Group+ and $1 : (\sqrt{5} - 1)/2 : (\sqrt{5} - 1)/2 : 1$ as Group−. After crossing the first EP $|\delta / \chi| = (\sqrt{5} - 1)/2$, the system enters the 1st-APT-B phase $(\sqrt{5} - 1)/2 < |\delta/\chi| < (\sqrt{5} + 1)/2$. The modes in Group+ are still degenerate and keep the fixed amplitude ratio, while the modes in Group− become nondegenerate with real eigenvalues $\lambda_{3,4} = \pm[\delta^2 + ((\sqrt{5} - 3)\chi^2/2)]^{1/2}$, exhibiting the fixed phase difference $[0, \pi/2, \pi, 3\pi/2]$. Tuning $\delta$ across the second EP $|\delta / \chi| = (\sqrt{5} + 1)/2$, the system enters the 2nd-APT-B phase. The modes in Group+ also become nondegenerate with the fixed phase difference $[0, \pi/2, 0, \pi/2]$ (see Supplementary Information X).

To verify the two-stage APT phase transition, we design three samples with different frequency detunings $\delta$. We take $r_2$ as 6, 6.4 and 6.7 mm respectively, while keeping $r_1 = 6$ mm. Their corresponding detunings $\omega_O\delta$ are extracted as 0, 0.035 and 0.06 GHz respectively and the indirect coupling $\omega_O\chi$ is the same as above. Therefore, the sample of ratio $|\delta / \chi| = 0$ corresponds to the APT-S phase, $|\delta / \chi| = 1.4$ in the 1st-APT-B phase and $|\delta / \chi| = 2.4$ in the 2nd-APT-B phase.

We demonstrate the two-stage APT symmetry breaking of the higher-order APT system with space-wave illuminations (see Supplementary Information V for the far-field experimental setup). The detected resonance frequencies are indicated with stars in Fig. 5b, which are consistent with the theoretical eigenvalues. The corresponding captured mode profiles are shown in Fig. 5c–j. Regarding the APT-S sample ($|\delta / \chi| = 0$) under normal incidence, Fig. 5c shows the low-Q mode $[1, (\sqrt{5} + 1)/2, (\sqrt{5} + 1)/2, 1]$ in Group + . For the samples $|\delta / \chi| = 1.4$ (1st-APT-B), modes in Group+ remain degenerate, but the degeneracy in Group− is lifted. Under normal incidence, we clearly identify the low-Q mode $[1, (\sqrt{5} + 1)e^{(-i\pi/3)}/2, (\sqrt{5} + 1)/2, e^{(-i\pi/3)}]$ in Group + (Fig. 5d), but cannot identify any mode in Group − . The reason is that normal incidence does not match the gradient phase difference $[0, \pi/2, \pi, 3\pi/2]$ in Group − , so that these modes cannot be efficiently excited. To address this issue, space waves are illuminated with a specific incidence angle $\theta_i = \arcsin[\lambda_i/4(d + 2R)]$ on the sample, where $\lambda_i$ denotes the wavelength of incident waves. The nondegenerate modes in Group− are clearly observed in Fig. 5 e, f. In the 2nd-APT-B region ($|\delta / \chi| = 2.4$), the high-frequency (low-frequency) modes are very close in spectra. To decern them, we also use normal and oblique incidences to efficiently excite them respectively (in Fig. 5g–j). These results confirm the scalability of our design, and selective excitation by harnessing the incident angle of space waves.

## Discussion

We proposed a radiative APT plasmonic design and demonstrated it with designer-plasmonic structures. Unlike previous photonic APT systems restricted in guided waves[26–29], our plasmonic APT systems are capable of harnessing spaces waves, due to the radiative property of leaky waves. Our system, exhibiting polarization-controlled APT phase transition, provides a platform to exploit the polarization of space waves with APT symmetry. The incidence angle of space waves also plays a role in decerning the higher-dimensional APT modes. Furthermore, attributed to the scalability, our system also provides a platform to demonstrate the higher-order APT phenomena, e.g. multi-stage APT phase transitions. Our work opens an avenue to construct APT-empowered radiative devices, and is promising in wireless sensing applications. Also, the subwavelength scale of our designer-plasmonic design is helpful in miniaturizing APT photonic devices. Our plasmonic APT design could be further extended to optical frequencies[53] (see Supplementary Information XI) and further interface with gain materials or topological structures[12,54], to spawn unusual lasers or intriguing non-Hermitian topological photonic phenomena.

## Methods

### Simulations

COMSOL Multiphysics is utilized to obtain the eigen-field patterns. The simulated near-field transmission spectra are obtained with CST Microwave Studio. The simulation model consists of two planar designer-plasmonic resonators with a distance $d = 10$ mm, encircled by the background horizontally. The groove-structured copper plates of 0.018 mm thickness stand on a $200 \times 80 \times 2$ mm dielectric substrate, with a copper coating on its bottom. The outer radius of the copper plate is $R = 12$ mm, and the groove number is $N = 60$. The metal is modeled as the perfect electrical conductor (PEC) in the microwave regime. The relative permittivity and loss tangent of the dielectric layer are 2.2 and 0.001 respectively. The near-field

excitation source is a discrete port, while plane-wave excitation is utilized in far-field simulation.

## Experiments

The sample is fabricated with the technology of printed circuit boards. The dielectric layer of the sample is made of F4BM with the relative permittivity $2.2 \pm 0.03$ and the loss tangent 0.001. Experimental measurements are carried out in the microwave anechoic chamber. The measurements of $S$ parameters are conducted with the vector network analyzer. Space waves are generated from a horn antenna.

## Data availability

The data generated in this study have been deposited in Figshare database under the following accession code. https://doi.org/10.6084/m9.figshare.21601923.v1.

## Code availability

The codes that support the plots within this paper are available from the corresponding authors upon reasonable request.

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

## Acknowledgements

The work at Zhejiang University was sponsored by the Key Research and Development Program of the Ministry of Science and Technology under Grants No. 2022YFA1404902 (F.G.), 2022YFA1404704 (H.C.), and 2022YFA1405200 (H.C.), the National Natural Science Foundation of China (NNSFC) under Grants No. 62171406 (F.G.), 61975176 (H.C.), 92163123 (Y.L.), No.11961141010 (H.C.), ZJNSF under Grant No. Z20F010018 (F.G.), National Key Laboratory Foundation No. 6142205200402 (F.G.), and the Fundamental Research Funds for the Central Universities.

## Author contributions

F.G. initiated the project. Y.Y., X.X. and Y.Z.L. conducted the theoretical analysis. Y.Y. did the simulations and experiments with the help of Z.Z., X.X. and Y.P. Y.Y. prepared the manuscript with inputs from other authors. C.W., E.L., Y.L. and H.C. participated in data collection and analysis. E.L., H.C., Y.L., and F.G. supervised the project.

## Competing interests
The authors declare no competing interests.
