## [Peer review file · Nature Communications]

REVIEWER COMMENTS

Reviewer #1 (Remarks to the Author):

In this work, the authors have implemented an experimental demonstration of anti-parity-time (anti-PT) symmetry by constructing two coupled designer surface-plasmonic resonators (DSPRs). Such a DSPR-based system allows the authors to realize the radiative anti-PT symmetry by utilizing the leaky waves to achieve the anti-Hermitian interaction, distinct from the Hermitian interaction associated with PT symmetry, the counterpart of anti-PT. Besides the observation of the essential features of anti-PT symmetry including its phase transition and supermode distributions, the system features the evolution of space waves after the anti-PT interaction owing to the particular wavelength range of the electromagnetic fields considered here.

After carefully going through the manuscript, although I found the work seems interesting, yet, I am not convinced that the reported findings are significant and meet the high standards of Nature Communications. In the following, I will list my main objections as well as the issues appearing in the current manuscript.

(1) The significance issue.

(1a) Although observing anti-PT symmetry in a new system will enrich the research on the subject, yet, only reporting its very basic features is not enough nor significant to result in new insights into the effect. Though one should acknowledge the authors' efforts on realizing radiative anti-PT symmetry using coupled DSPRs with different radii, the severe drawback of the work is that no new effect or phenomenon is revealed from the designed system. In other words, anti-PT phase transition and its supermodes evolution (or field distributions) have already been reported and demonstrated in the past in different systems. Without the disclosure of new physics, the reported work is simply another demonstration of the anti-PT effect.

(1b) On the other hand, the claimed feature of "space waves" is not surprising at all. In fact, this feature is natural to the wavelength of the electromagnetic field studied here. Moreover, their design mechanism is difficult to be extended to visible light as they stated in the Discussion.

(1c) The designed system has limited functionalities. As the authors stated, in order to observe even the anti-PT phase transition, they had to design a set of DSPRs with different radii. In other words, one DSPR only generates one data point in the phase-transition diagram. This drawback alternatively puts a limit on most potential applications, especially the claimed sensing performance.

(2) Many incorrect or unsupported statements.

(2a) The authors gave an incorrect introduction on the history of anti-PT symmetry. As noted in the introduction, the authors quoted reference [20] as the first work of anti-PT symmetry [21]. In fact, reference [20] has nothing to do with anti-PT symmetry, as it didn't show the anti-commutation relation, nor the imaginary interaction, as well as other associated features.

(2b) In lines 130-132, the statement on the indirect coupling is incorrect. In fact, such an indirect coupling realized in previous photonic experiments (e.g., Ref. [21]; ACS Photonics 7, 3035 (2020); Nat. Commun. 12, 486 (2021); PRL 123, 193604 (2019)) is also a linear coupling mechanism.

(2c) In lines 205-209, the authors interpreted the discrepancies between theory and experiment as the missing Jordan vector. This interpretation is too mathematical other than physical. We strongly suggest the authors to use physical effects to explain rather than a mathematical term. The same issue also appears couple of times in other places of the main text.

(2c) In lines 250-252, the comparison only with the work [23] but ignoring other highly-relevant experiments mentioned above is misleading and incomplete.

(2d) The employed "leaky waves" for the realization of imaginary coupling is not new. This is similar to atomic coherence leakage utilized in the first work on anti-PT symmetry [21].

(2e) The authors claimed their system may be useful for sensing applications. However, this claim is lack of supports. Considering the explored wavelength in this work, in fact, the current system won't display appreciable advantages in designing exceptional-point sensors in terms of sensitivity and versatility.

(3) Other issues.

(3a) The authors used "t" to represent the thickness of the substrate but also used it for the transmission coefficient. Please use different symbols to distinguish these two different quantities by avoiding confusion.

(3b) The math equations in lines 121 and 123 are incorrect in dimension. Please consider to correct them.

(3c) In line 124, what "proper regions" should be? Please provide a quantitative estimation if possible.

(3d) It is unclear what advantages one can obtain with the space waves emphasized in this work.

(3e) Reference issues. A few anti-PT experimental works have been overlooked by the authors. As they are highly relevant to the current work, we strongly encourage the authors to give them credits.

In brief, as no new features or insights are available from the current work, I am not convinced that the manuscript meets the high standards of the journal. Moreover, the anti-PT features have been well understood in terms of phase transition and supermode dynamics. As such, if only with the current findings, the manuscript is more suitable for a specialized technical journal such as Optics Express and Physical Review Applied.

Reviewer #2 (Remarks to the Author):

Anti-parity-time (APT) symmetry is an emerging non-Hermitian mechanism, which has recently attracted intense interests from fundamental physics to technological applications. The authors of this manuscript propose a leaky-wave scheme for constructing radiative APT systems, and demonstrate it using designer-plasmonic structures. Conceptually, this work contributes to the photonic community a new concept i.e. radiative APT plasmonics, fundamentally different from conventional photonic APT systems for guided waves. Technologically, the leaky-wave scheme enables the realization of APT symmetry in the linear optics region, thus breaking the restriction of strong pumping light as previous photonic realizations based on nonlinear couplings. Moreover, the paper is well organized based on convincing results, covering theoretical analysis, numerical simulations, and experimental verifications. Overall, I think this work has the standards that one may expect for a publication in Nature Communications. I strongly recommend accepting this manuscript after addressing some minor concerns.

Minor concerns:

1. It is impressive that the 2-mm thicknesses of the samples are much smaller than the operational wavelengths. I am wondering whether that thickness achieves the limit.
2. The corresponding frequency of the resonance mode in Fig. 2b should be given.
3. The field patterns of photonic APT systems are firstly captured in experiments, therefore showing the experimental setup would be helpful for reproducing.
4. The references should be carefully selected, for example, [Phys. Rev. Lett. 123(19), 193604 (2019)] should be related to atomic APT systems and [Nat. Commun. 12(1), 486 (2021)] should be related to photonic APT systems. It is also suggested to cite [Chinese Phys. B 31(1), 014215 (2022)] when referring to enhanced Sagnac effect in APT systems.
5. At line 49, the full name of APT should be given.
6. Some of the references are missing page numbers, such as Ref. [9] at line 368, Ref. [15] at line 379, Ref. [20] at 388-389, and Ref. [25] at 397-398. Please check all the references carefully to avoid such typos.

Response Letter

We are grateful for the constructive comments on the manuscript from the referees, which help us better communicate the novelty and significance of our work.

According to the referees' suggestions, we have added extra experiments and data to strengthen the significance and clarity of our work. In the text below, the referees' comments are quoted in *italics* and followed by our detailed response (in blue). The corresponding updates in our manuscript and supplementary materials are highlighted in red, and these updates are also quoted in red in the following text.

General comments from Referee #1:

In this work, the authors have implemented an experimental demonstration of anti-parity-time (anti-PT) symmetry by constructing two coupled designer surface-plasmonic resonators (DSPRs). Such a DSPR-based system allows the authors to realize the radiative anti-PT symmetry by utilizing the leaky waves to achieve the anti-Hermitian interaction, distinct from the Hermitian interaction associated with PT symmetry, the counterpart of anti-PT. Besides the observation of the essential features of anti-PT symmetry including its phase transition and supermode distributions, the system features the evolution of space waves after the anti-PT interaction owing to the particular wavelength range of the electromagnetic fields considered here.

After carefully going through the manuscript, although I found the work seems interesting, yet, I am not convinced that the reported findings are significant and meet the high standards of Nature Communications. In the following, I will list my main objections as well as the issues appearing in the current manuscript.

Response from Authors:

We thank the referee for noticing the unique feature of our system, i.e. “the evolution of space waves after the anti-PT interaction”, and considering our work “interesting”.

We also fully understand the concern about the significance of our work from the referee. We address it by further exploring the degrees of freedom (DoFs) of space waves, and revealing new APT physics and phenomena. The point-by-point responses are provided as below.

Specific comments from Referee #1:

Referee #1 Comment 1:

(1) The significance issue.

(1a) Although observing anti-PT symmetry in a new system will enrich the research on the subject, yet, only reporting its very basic features is not enough nor significant to result in new insights into the effect. Though one should acknowledge the authors' efforts in realizing radiative anti-PT symmetry using coupled DSPRs with different radii, the severe drawback of the work is that no new effect or phenomenon is revealed from the designed system. In other

words, *anti-PT phase transition and its supermodes evolution (or field distributions) have already been reported and demonstrated in the past in different systems. Without the disclosure of new physics, the reported work is simply another demonstration of the anti-PT effect.*

Response from Authors:

We appreciate the referee's encouraging words "*observing anti-PT symmetry in a new system will enrich the research on the subject*", as well as the concern about the significance of our work. To reveal the uniqueness of our system, i.e. the radiative property, we show two more new exotic APT phenomena, by exploring the far-field DoFs of space waves (polarizations and incident angles).

New phenomenon #1: Polarization-controlled APT phase transition

The first new phenomenon is the polarization response of the radiative plasmonic APT system. When the polarization of the incident plane wave is changed from horizontal (H) to vertical (V), the manifestation of the illuminated APT system switches from APT symmetric to APT symmetry-broken phase (in Fig. R1.1).

Fig. R1.1. The polarization-controlled APT phase transition. (a) The evolution of eigen frequencies as the detuning $r_R - r_L$ changes for odd (H-polarized) and even (V-polarized) modes. The stars mark the detected resonance frequencies on the sample $r_R - r_L = 0.27$ mm. (b) The far-field excitation setup. The red and blue arrows denote horizontally and vertically polarized space waves, respectively. Points A and B denote the probe positions for odd and even modes respectively. (c) The experimental transmission spectra under illuminations of H-polarized (red) and V-polarized (blue) plane waves, respectively. The stars mark the resonant-peak positions. (d-f) The field patterns E_z under H-polarized plane-wave illuminations at 3.725 GHz (d), and V-polarized plane-wave illuminations at 3.72 GHz (e) and 3.76 GHz (f). The distributions of E_z and $|E_z|$ are shown with rainbow and hot colors, respectively.

This exotic phenomenon is attributed to the mechanism that leaky waves link the H and V polarizations of space waves to the odd and even parities of DSPR modes. The parities of DSPR modes further modulate the leaky-wave couplings (in Fig. R1.2), resulting in different indirect coupling strengths for odd- and even-mode APT systems (see page 6 in the revised manuscript for a detailed analysis). Due to the different coupling strengths, the APT system is in the different APT phases for odd and even modes in the blue region of Fig. R1.1a, Thus, we can induce the APT phase transition by switching the polarizations of space waves.

Fig. R1.2. The mode-parity modulation on leaky waves.

Regarding this phenomenon, we have added the following part on page 9 and included Fig. R1.1 as Fig. 4 in the revised manuscript:

We further investigate the polarization responses of the plasmonic APT system, since the leaky wave links the polarization of space waves to the parity of DSPR modes. Specifically, the horizontal (H) and vertical (V) polarizations are linked to the odd- and even-parity modes, respectively. The even and odd modes exhibit different indirect coupling strengths due to their parity modulations on leaky waves (in Fig. 2f). Therefore, the APT system exhibits two sets of APT phase transitions for odd and even modes (in Fig. 4a). In the blue region of Fig. 4a, the APT system is in the APT-S phase for odd modes while in the APT-B phase for even modes. Since the links between the space-wave polarizations and DSPR-mode parities, we switch the polarizations of illuminating space waves (in Fig. 4b) to induce the APT phase transition. We design the sample ($r_R = 6.27$ mm and $r_L = 6$ mm) in the blue region. A vertical near-field probe is utilized to capture the E_z components of field patterns. We probe the fields at positions A and B (in Fig. 4b) under H- and V- polarized incidence, respectively. When we rotate the space-wave polarization from H to V, the number of resonance peaks changes from one to two (shown in Fig. 4c). The spectral results indicate the polarization-controlled phase transition from APT-S to APT-B, consistent with the theoretical results in Fig. 4a.

The polarization-controlled APT phase transition is further verified by the captured field patterns at the resonance peaks. Under H-polarized incidence, the pattern shows the low-Q mode $[1, e^{i\pi/3}]$ of odd parity at 3.725GHz, thus confirming the APT-S phase. While under the V-polarized incidence, the captured patterns, corresponding to the even parity, show $[2, -i]$ at 3.72 GHz and $[1, -2i]$ at 3.76 GHz, respectively, thus confirming the APT-B phase.

To the best of our knowledge, the polarization-controlled APT phase transition has not been realized before. This phenomenon confirms the capability of our radiative plasmonic

APT system in harnessing the far-field DoFs of space waves. This capability is promising in constructing APT-empowered radiative devices and wireless sensing applications.

New phenomenon #2: Two-stage phase transition in higher-order plasmonic APT systems.

The second new phenomenon is the multi-stage APT phase transition in higher-order APT systems (in Fig. R1.3), observed by tuning incidence angles of space waves. This phenomenon is based on the scalability of our radiative plasmonic system. Since the in-plane propagation component of leaky waves enables the imaginary coupling between two spatially separated plasmonic resonators, the design mechanism can be further extended to higher-order systems. Moreover, the incoming direction of space waves plays a unique role in identifying high-order APT supermodes.

Fig. R1.3. Observation of multi-stage APT phase transitions under space-wave illuminations. (a) The top view of the higher-order APT-symmetry system. From left to right, the inner radii of the four DSPRs are $r_1, r_2, r_1,$ and $r_2,$ respectively. (b) The real (black) and imaginary part (white) of the calculated eigenvalues of the higher-order system versus the detuning $r_2 - r_1$. The blue, purple and pink regions denote the APT-S, 1st-APT-B and 2nd-APT-B phases, respectively. The red dots represent the EPs. Stars mark the experimentally detected resonance frequencies at $r_2 - r_1 = 0, 0.4,$ and 0.7 mm. The inset is the schematic of normal (red) and oblique (blue) illuminations to excite modes in Group+ and Group-, respectively. (c-j) The measured field patterns of APT-S at $|\delta/\kappa| = 0$ (c), 1st-APT-B at $|\delta/\kappa| = 1.4$ (d-f), and 2nd-APT-B phases at $|\delta/\kappa| = 2.4$ (g-j), respectively. The distributions of E_z and $|E_z|$ are shown with rainbow and hot colors, respectively.

Regarding this phenomenon, we have added the following part on pages 10-11 and included Fig. R1.3 as Fig. 5 in the revised manuscript:

Fig.5(a) shows the higher-order APT system, which consists of four DSPRs with inner radii [r_1, r_2, r_1, r_2], while other parameters remain the same as the above structures. In the basis $[\psi_1^{nf}, \psi_2^{nf}, \psi_3^{nf}, \psi_4^{nf}]^T$, the Hamiltonian of this system is expressed as $\hat{H} = \omega_0 + i\gamma_0 + \omega_0 \hat{H}_4$.

$$\hat{H}_4 = \begin{bmatrix} -\delta & i\chi & 0 & 0 \\ i\chi & \delta & i\chi & 0 \\ 0 & i\chi & -\delta & i\chi \\ 0 & 0 & i\chi & \delta \end{bmatrix}$$

The fourth-order effective Hamiltonian \hat{H}_4 obeys the four-dimensional anti-commutation relation $\{\hat{H}_4, P_4 T\} = 0$, where the four-dimensional parity operator is $P_4 = [0,0,0,1; 0,0,1,0; 0,1,0,0; 1,0,0,0]$.

During sweeping the frequency detuning δ , the higher-order APT system undergoes two-stage APT-symmetry breakings (in Fig. 5b), marked by the two distinct EPs ($|\delta/\chi| = (\sqrt{5} \mp 1)/2$). When $|\delta/\chi| < (\sqrt{5} - 1)/2$, four degenerate modes differ with the imaginary parts of their eigenvalues $\lambda_{1,2,3,4} = \pm i \sqrt{\frac{-2\delta^2 + (\pm\sqrt{5}+3)\chi^2}{2}}$. We term this region as the APT-S phase, where all four degenerate modes can be classified into two groups according to their fixed amplitude ratios, i.e. $|\psi_1^{nf}| : |\psi_2^{nf}| : |\psi_3^{nf}| : |\psi_4^{nf}| = 1 : \frac{1+\sqrt{5}}{2} : \frac{1+\sqrt{5}}{2} : 1$ as Group+ and $1 : \frac{\sqrt{5}-1}{2} : \frac{\sqrt{5}-1}{2} : 1$ as Group-. After crossing the first EP $|\delta/\chi| = (\sqrt{5} - 1)/2$, the system enters the 1st-APT-B phase $(\sqrt{5} - 1)/2 < |\delta/\chi| < (1 + \sqrt{5})/2$. The modes in Group+ are still degenerate and keep the fixed amplitude ratio, while the modes in Group- become nondegenerate with real eigenvalues $\lambda_{3,4} = \pm \sqrt{\frac{2\delta^2 + (\sqrt{5}-3)\chi^2}{2}}$, exhibiting the fixed phase difference $[0, \frac{\pi}{2}, \pi, \frac{3\pi}{2}]$. Tuning δ across the second EP $|\delta/\chi| = (1 + \sqrt{5})/2$, the system enters the 2nd-APT-B phase. The modes in Group+ also become nondegenerate with the fixed phase difference $[0, \frac{\pi}{2}, 0, \frac{\pi}{2}]$ (SMX for detailed analysis of eigenmodes).

To verify the two-stage APT phase transition, we design three samples with different frequency detunings δ . We take r_2 as 6, 6.4 and 6.7 mm respectively, while keeping $r_1 = 6$ mm. Their corresponding detunings $\omega_0\delta$ are extracted as 0, 0.035 and 0.06 GHz respectively and the indirect coupling $\omega_0\chi$ is the same as above. Therefore, the sample of ratio $|\delta/\chi| = 0$ corresponds to the APT-S phase, $|\delta/\chi| = 1.4$ in the 1st-APT-B phase and $|\delta/\chi| = 2.4$ in the 2nd-APT-B phase.

We demonstrate the two-stage APT symmetry breaking of the higher-order APT system with space-wave illuminations (SMV for the far-field experimental setup). The detected resonance frequencies are indicated with stars in Fig. 5b, which are consistent with the theoretical eigenvalues. The corresponding captured mode profiles are shown in Fig. 5c-j. Regarding the APT-S sample ($|\delta/\chi| = 0$) under normal incidence, Fig. 5c shows the low-Q mode $[1, \frac{1+\sqrt{5}}{2}, \frac{1+\sqrt{5}}{2}, 1]$ in Group+. For the samples $|\delta/\chi| = 1.4$ (1st-APT-B), modes in Group+ remain degenerate, but the degeneracy in Group- is lifted. Under normal incidence, we clearly identify the low-Q mode $[1, \frac{1+\sqrt{5}}{2} e^{-i\pi/3}, \frac{1+\sqrt{5}}{2} e^{-i\pi/3}]$ in Group+ (Fig. 5d), but cannot identify any mode in Group-. The reason is that normal incidence does not match the gradient

phase difference $\left[0, \frac{\pi}{2}, \pi, \frac{3\pi}{2}\right]$ in Group-, so that these modes cannot be efficiently excited. To address this issue, space waves are illuminated with a specific incidence angle $\theta_i = \arcsin [\lambda_i / 4(d+2R)]$ on the sample, where λ_i denotes the wavelength of incident waves. The nondegenerate modes in Group- are clearly observed in Fig5. e-f. In the 2nd-APT-B region ($|\delta/\lambda| = 2.4$), the high-frequency (low-frequency) modes are very close in spectra. To discern them, we also use normal and oblique incidences to efficiently excite them respectively (in Fig. 5g-j). These results confirm the scalability of our design, and selective excitation by harnessing the incident angle of space waves.

To the best of our knowledge, this is the first observation of the multi-stage APT phase transition. Moreover, this scalability paves the way to study the interaction between APT symmetry and more complex crystal symmetries, e.g. Su-Schrieffer-Heeger (SSH) model. Therefore, our approach can provide platforms to investigate more rich physics, such as APT topological phenomena.

Referee #1 Comment 2:

(1b) On the other hand, the claimed feature of "space waves" is not surprising at all. In fact, this feature is natural to the wavelength of the electromagnetic field studied here. Moreover, their design mechanism is difficult to be extended to visible light as they stated in the Discussion.

Response from Authors:

We thank the referee for the critical comment. The referee considers that “*this feature (i.e. the space wave) is natural to the wavelength of the electromagnetic field studied here*”, thus concluding that “*space waves*” is not surprising at all.

Actually, the space wave feature of our radiative system is determined by the unique design of the structure, instead of the working wavelength. Generally, since the golden Maxwell equations are scale-free, it does not limit the working modes (i.e. “space wave” or “guided wave”) to specific wavelengths. For instance, antennas working for space waves, and waveguides working for guided waves are both applicable to the same wavelength. The crucial difference between an antenna and a waveguide is their distinct structural designs. Similarly, realizing the radiative APT is NOT *natural* to the wavelength in our work, while depends on our unique design. In addition, our design scheme can be extended to optical frequencies, since the designer plasmon (also known as spoof plasmon) in our system is the low-frequency analogue of surface plasmon in optical frequencies.

Thus, we would like to clarify in the following part from three aspects, i.e. (i) the unique design of radiative systems. (ii) the approach of extension to optical frequencies. (iii) the uniqueness in harnessing DoFs of space waves

1. The unique design of radiative APT systems

To the best of our knowledge, our work is the first realization of a radiative APT system based on the plasmonic structure and leaky-wave coupling. As a counter example, the previous study

[e.g. *Light Sci. Appl.* 8, 88 (2019)] on APT symmetry, which is only applicable to guided waves, is also in the microwave frequency range. That structure (in Fig. R1.4) consists of dielectric waveguides and lossy coupling elements. The different configuration further confirms that APT structure responding to space waves needs specific designs, and is NOT *natural* to the microwave wavelength.

Fig. R1.4 | Structure comparison between guided and radiative APT systems. (a) The schematic of the photonic APT system for guide modes [*Light Sci. Appl.* 8, 88 (2019)]. (b) The top view of our radiative APT system.

Here we take the substrate thickness to elucidate the unique design of our system. The thickness h of the dielectric substrate is elaborately designed to ensure efficient imaginary coupling by leaky waves. The strength of imaginary coupling decreases as h decreases, according to the theoretical expression $i\chi \propto h \cdot \int \frac{1}{|C_{NF}| \cdot |C_{FF}|} \psi_{1(2)}^{nf} \cdot \psi_{2(1)}^p dx dy$ (see page 5 of the manuscript for detailed analysis). This relation is also verified with simulations (in Fig. R1.5). Thus, when the substrate is too thin, the leaky wave coupling will tend to vanish. In this scenario, the new system is composed of two independent high-Q resonators, and does not show any APT features. Also, the new system hardly interacts with space waves.

Fig. R1.5. The imaginary coupling strength χ as the thickness h of the substrate evolves.

Regarding this design, we added this part on page 22 in the supplementary note XII and included Fig. R1.5 as Fig. S9.

2. The approach of extension to optical frequencies

Our design mechanism of APT systems is not limited to microwave frequencies. It can be further extended to optical frequencies (in Fig. R1.6), and the extension approach lies in the following two aspects:

(i). The leaky-wave mechanism extension

The system (dielectric on metal) can host surface EM waves in both microwave and optical frequencies, and also host corresponding leaky waves above the light line. The only difference is the metal properties in the two frequency regions. In microwave, metal approximately behaves as a perfect electric conductor (PEC), while in optics it follows the dispersion of the Drude model. Such a difference manifests as that the leaky waves in optic frequencies penetrate metal deeper than that in the microwave. Despite the existence of such a difference, leaky waves exist in both frequency regions.

(ii). The plasmonic resonator extension

In optical frequency, plasmonic-resonator extension is also feasible. Since the designer surface plasmon resonators (DSPRs) are low-frequency analogous to the metallic nanodisks at optical frequencies [Rev. Mod. Phys. **94**, 025004 (2022)]. To achieve the extension, we replace the textured metal disk in the microwave with metallic nanodisks in optics.

Conclusively, with the same configuration and mechanism, our designed architecture can be extended to optical frequencies. The extension is also feasible in the experimental realization [Science **373**, 1133-1137 (2021)].

Fig. R1.6. The extension of APT design to the visible frequency. (a) The top and the front view of the plasmonic APT system. It consists of two silver (Ag) nanodisks with a distance $d = 170$ nm. The radii of the two nanodisks are r_R (right) and r_L (left), and the thickness h is 20 nm. The background structure is composed of two layers. The top layer is a dielectric layer ($\epsilon_r = 1.5$)

of 50 nm thickness. The bottom layer is a silver substrate of 20 nm thickness. (b) The simulated evolution of the transmission spectra as $r_L - r_R$ varies. The inset shows the simulation setup. (c-d) The field patterns E_z correspond to the APT-S phase ($r_L - r_R = 0$). (e-f) The field patterns E_z correspond to the APT-B phase ($r_L - r_R = 3.5$ nm). The distributions of E_z and $|E_z|$ are shown with rainbow and hot colors.

Regarding the extension to optical frequency, we have added the following part on pages 21-22 in the supplementary material XI and included Fig. R1.6 as Fig. S8:

As shown in Fig. S8a, the APT system consists of two silver (Ag) nanodisks with the dielectric layer and metal substrate (Ag) at the bottom. The nanodisk also supports a tightly confined dipole mode and the horizontal decay length is $L_x = 70$ nm from the edge of the disk. The background structure hosts horizontally propagating waves, which radiate into free space by the out-of-plane leakage, providing the indirect coupling channels. We set the edge-edge distance of the two disks sufficiently distant as $d = 170$ nm, which is larger than $2L_x$.

To verify the symmetry-breaking process, we design samples with different frequency detunings. We change the inner radius r_L of the left DSPR as 60, 57.5, and 56.5 nm respectively, while keeping the $r_R = 60$ nm for the right DSPR.

The simulated spectra are shown in Fig. S8b. When $r_L = 60$ nm, the frequency detuning parameter $\omega_0\delta$ is extracted as 0, which corresponds to the APT-S phase. Thus, the simulated transmission spectrum only has one peak. The low-Q eigenmode [1, 1] and the high-Q eigenmode [1, -1] are excited by the far-field (Fig. S8c) and near-field (Fig. S8d) source, respectively. Due to the effect of the near-field source, the amplitude of the high-Q eigenmode slightly deviates from the calculated result. When $r_L = 56.5$ nm, the simulated transmission spectrum splits, which corresponds to the APT-B phase. The low-frequency eigenmode (Fig. S8e) and high-frequency eigenmode (Fig. S8f) have the same phase difference $-\pi/2$, which is consistent with the theoretical analysis.

We are grateful for the referee's constructive suggestion, which stimulates us to validate the extension capability of our design, enhancing the significance of our work. To emphasize the generality and extension capability of our approach, we changed the title to "**Radiative Anti-Parity-Time Plasmonics**" in the revised manuscript, rather than the specific designer-plasmonic structures.

3. The uniqueness in harnessing DoFs of space waves

Actually, the feature of space waves is *surprising*, since it also enables us to harness the far-field DoFs for controlling or selectively exciting APT supermodes. It has been shown in response to comment 1a.

Referee #1 Comment 3:

(1c) The designed system has limited functionalities. As the authors stated, in order to observe even the anti-PT phase transition, they had to design a set of DSPRs with different radii. In

other words, one DSPR only generates one data point in the phase-transition diagram. This drawback alternatively puts a limit on most potential applications, especially the claimed sensing performance.

Response from Authors:

We thank the reviewer for the concern about the functionalities of our APT system. It is worth noting that we use different samples here just for the convenience of experimental characterizations, but not because that is the only tuning approach. For instance, another tuning approach, i.e. changing the environmental index, is also applicable to our system. With such tuning, we can induce the APT phase transition without using different samples, thus promising applications in sensing applications.

1. Near-field detection of environmental variations

Since our plasmonic APT system is an open system, the phase transition can be achieved by changing environmental factors, e.g. permittivity (in Fig. R1.7).

Fig. R1.7. Detection of environmental variations with the plasmonic APT system. (a) The APT system for environmental variation detection. The inner radius of the left and right DSPRs is 6 mm. The left DSPR is covered with the media of the relative permittivity ϵ_r . (b) The evolution of the simulated transmission spectra as the permittivity ϵ_r changes.

Regarding this sensing application, we have added the following part on page 18 in the supplementary note VIII and included Fig. R1.7 as Fig. S6:

We consider the case that the left DSPR is covered by the media with a relative permittivity ϵ_r , and keep the other parts of the system unchanged (shown in Fig. S6a). When the permittivity of the environmental media changes, the resonant frequency of the left DSPR changes, further leading to the change of the frequency detuning. It means that the APT-phase transition can be realized by changing the permittivity of the environmental media.

Based on the above analysis, we simulate the near-field transmission spectra as shown in Fig.

S6b when $\epsilon_r = 1, 1.5, 2$ and 3 . From the figure, the resonant peak of the transmission spectra splits with the increase of ϵ_r , which means the system evolves from the APT-S phase to APT-B phase. Therefore, we can detect the environmental variations according to the resonance-frequency split of the APT system.

2. Remote sensing of environmental variations

The radiative feature of our system also enables us to observe the APT phase transition in the far field by using the same tuning approach as above (shown in Fig. R1.8). **In other words, we can remotely sense the changing of the environmental index, using the radiative APT systems.** This feature confirms that our radiative system is promising in constructing APT-empowered radiative devices and remote sensing applications.

Fig. R1.8. Remote sensing of environmental variations. (a) The simulation setup for remote sensing of environmental variations. (b) The evolution of the simulated reflection spectra as the media permittivity ϵ_r changes.

Regarding this sensing application, we have added the following part on page 19 in the supplementary note IX and included Fig. R1.8 as Fig. S7:

We use the same sample and tuning approach here as in the near-field sensing (SMVIII for details). The sample is illuminated with H-polarized space waves, and we use reflection spectra to observe the manifestation of the radiative APT system (shown in Fig. S7a).

The simulated reflection spectra are shown in Fig. S7b when $\epsilon_r = 1, 2$ and 3 . From the figure, the resonant peak of the transmission spectra splits with the increase of ϵ_r , which means the system evolves from the APT-S phase to the APT-B phase. Therefore, we can remotely sense the environmental variations according to the reflection spectra of our radiative APT system. These results confirm that our radiative system is promising in constructing APT-empowered radiative devices and remote sensing applications.

Referee #1 Comment 4:

(2) Many incorrect or unsupported statements.

(2a) The authors gave an incorrect introduction on the history of anti-PT symmetry. As noted in the introduction, the authors quoted reference [20] as the first work of anti-PT symmetry [21]. In fact, reference [20] has nothing to do with anti-PT symmetry, as it didn't show the anti-commutation relation, nor the imaginary interaction, as well as other associated features.

Response from Authors:

We thank the referee for pointing out the reference issue.

We have checked all the references about APT symmetry carefully. Undoubtedly, Ref. [21] (*Nature Physics* 12, 1139–1145 (2016)) is the first experimental work about APT symmetry, which is widely recognized by the community (e.g., Ref. [22]; Ref. [33]). However, Ref. [20] (*Phys. Rev. A* 88, 053810 (2013)) is also considered as an effort to APT symmetry by some photonic peers (e.g., Ref. [25]; Ref. [26]; Ref. [30]). To avoid controversy, we cite Ref. [20] in the photonic APT works of the introduction part. We have revised the introduction as below, and the corresponding references have also been revised in the reference part of the manuscript.

On page 3: Subsequently, anti-parity-time (APT) symmetry has also been proposed and aroused intense interests across multi disciplines from atomics²⁰⁻²³, photonics²⁴⁻³⁰, classical³¹ and quantum³² circuits, thermology³³ to magnetics³⁴, etc.

Referee #1 Comment 5:

(2b) In lines 130-132, the statement on the indirect coupling is incorrect. In fact, such an indirect coupling realized in previous photonic experiments (e.g., Ref. [21]; *ACS Photonics* 7, 3035 (2020); *Nat. Commun.* 12, 486 (2021); *PRL* 123, 193604 (2019)) is also a linear coupling mechanism.

Response from Authors:

We thank the referees for pointing out the issue. This is a misunderstanding caused by the vague statement "linear coupling mechanism". What we emphasized here is that the leaky-wave coupling in our plasmonic APT system is a mechanism of **linear optics**. To express more rigorously, we revised the statement on page 6 in the manuscript as below:

It is noteworthy that such photonic indirect coupling, resulting in a plasmonic APT system in the spectral domain, is a mechanism of linear optics, rather than nonlinear optics which usually require high-power input.^{21, 26, 29}

To show the uniqueness of our work more clearly, we also compared the features of our work and those in the references mentioned in the comment in the following table (the different features are marked in red):

	Ref. [21]	ACS Photonics 7, 3035 (2020)	Nat. Commun. 12, 486 (2021)	PRL 123, 193604 (2019)	Our work
Type	Atomic	Photonic APT	Photonic APT	Photonic APT	Photonic

	APT				APT
Domain	Spectral domain	Spatial domain	Spectral domain	Spectral domain	Spectral domain
Coupling mechanism	Atomics	Linear optics	Nonlinear optics	Nonlinear optics	Linear optics

Referee #1 Comment 6:

(2c) In lines 205-209, the authors interpreted the discrepancies between theory and experiment as the missing Jordan vector. This interpretation is too mathematical other than physical. We strongly suggest the authors to use physical effects to explain rather than a mathematical term. The same issue also appears couple of times in other places of the main text.

Response from Authors:

We thank the referees for the constructive suggestion. Since the missing Jordan vector refers to the missing dimension at EP, we have revised the interpretation of the deviation with the emergence of the missing eigenstate. Such missing eigenstate emerges when the APT system is driven in the presence of background loss, thus causing the deviation of experimental field patterns from theoretical results (detailed analysis is shown in supplementary note VI). We have revised the manuscript on page 8 as below:

Such deviation is due to the emergence of the missing eigenstate $[1, i]^T$ in the presence of the background loss γ .

Referee #1 Comment 7:

(2c) In lines 250-252, the comparison only with the work [23] but ignoring other highly-relevant experiments mentioned above is misleading and incomplete.

Response from Authors:

We thank the referees for reminding us of the incomplete comparison. To clarify the significance of our work, we have removed this misleading statement and added the following part on page 12 in the discussion of the manuscript:

Unlike previous photonic APT systems restricted in guided waves²⁶⁻²⁹, our plasmonic APT systems are capable of harnessing spaces waves, due to the radiative property of leaky waves. Our system, exhibiting polarization-controlled APT phase transition, provides a platform to exploit the polarization of space waves with APT symmetry. The incidence angle of space waves also plays a role in decerning the higher-dimensional APT modes.

Referee #1 Comment 8:

(2d) The employed "leaky waves" for the realization of imaginary coupling is not new. This is similar to atomic coherence leakage utilized in the first work on anti-PT symmetry [21].

Response from Authors:

We thank the referee for the critical comment, which is about the uniqueness of the "leaky wave"

compared with "*atomic coherence leakage*". The leaky wave (in our work) is fundamentally different from "atomic coherence leakage" (Fig. R1.9 a). Specifically, the leakage of leaky waves refers to the electromagnetic (EM) wave leaking from the in-plane guided modes. While the leakage of "*atomic coherence leakage*" refers to the *flying atoms* escaping from the atomic spin waves. We also compared them in detail in the following two aspects.

(1) Different physical properties

Firstly, we compared their physical properties in the following table:

	Atomic coherence leakage (Ref. [21])	Leaky wave
Type	Atoms (quantum wave)	EM wave (classical wave)
Non-Hermitian mechanism	Randomness & irreversibility	Out-of-plane leakage
Property	N.A.	1. Polarization response to space wave (Fig. R1.1) (Out-of-plane radiation component) 2. Parity modulation (Fig. R1.2) (In-plane radiation component)

We further elucidate the uniqueness of leaky waves with its concept. It is a type of special radiative EM mode derived from guided modes without radiation leakage [Kong, J. A. Electromagnetic Wave Theory. 419 (2008)]. Fig. R1.9 b-c show the typical structure and field pattern of leaky waves. The typical structure consists of a dielectric layer coated with the conductor (in Fig. R1.9 b). Such the structure can host guided modes without radiation leakage, which propagate along the z direction and decay in the air along the x direction. Thus, its wavenumber is obtained as $\vec{k}_{NR} = \hat{z}\beta_{nz} + \hat{x}i\alpha_{nx}$, where β_{nz} and α_{nx} denote the propagation constant and attenuation rate, respectively. Small perturbations can derive these non-radiative guided modes into radiative leaky modes, which radiate into the air along the x direction. The wavenumber of the radiative leaky mode is $\vec{k}_R = \hat{z}(\beta_z + i\alpha_z) + \hat{x}(\beta_x - i\alpha_x)$, which satisfies the dispersion relation $(\omega/c)^2 = (\beta_z + i\alpha_z)^2 + (\beta_x - i\alpha_x)^2$. The corresponding field pattern is shown in Fig. R1.9 c.

Therefore, the unique origin and radiative feature make the leaky wave fundamentally different from the atomic coherence leakage.

Fig. R1.9. Different physical properties. (a) Schematics of the atomic coherence leakage and leaky wave. (b) The typical structure of leaky waves. (c) The field pattern of the leaky wave.

(2) Different features of APT systems

The APT systems based on these two mechanisms have different properties in **architecture**. Our leaky-wave-enabled APT system is a passive system, while the atomic one requires pumping light to dress atoms.

Overall, our leaky-wave approach is totally different from the atomic coherence leakage. To the best of our knowledge, it is the first time that the leaky wave is explored to realize APT systems.

Referee #1 Comment 9:

(2e) The authors claimed their system may be useful for sensing applications. However, this claim is lack of supports. Considering the explored wavelength in this work, in fact, the current system won't display appreciable advantages in designing exceptional-point sensors in terms of sensitivity and versatility.

Response from Authors:

We thank the referees for the critical comment. Regarding the sensing applications, we have added supporting data in two aspects based on our APT system, i.e. (i) sensing environmental variations. (ii) extension to optical frequencies.

(1) Sensing environmental variations

Since our plasmonic APT system is an open system, the phase transition can be achieved by changing environmental factors, e.g. permittivity. Thus, the open APT system can be utilized to **sense environmental variations**. Detailed information about the sensing application has been shown in response to comment 1c.

The radiative feature of our system also enables us to observe the APT phase transition in the far field by using the same tuning approach as above. In other words, we can **remotely sense the changing of the environmental index**, using the radiative APT systems. This feature

confirms that our radiative system is promising in constructing APT-empowered radiative devices and remote sensing applications. The numerical results are included in the response to comment 1c.

(2) Extension to optical frequencies

Our radiative APT design is not restricted to *the explored wavelength in this work*. What we proposed is a general design approach, which can be extended into optical frequencies. To validate the extension capability of our design, the extension mechanism and numerical results are included in the response to comment 1b.

Referee #1 Comment 10:

(3) Other issues.

(3a) The authors used "t" to represent the thickness of the substrate but also used it for the transmission coefficient. Please use different symbols to distinguish these two different quantities by avoiding confusion.

Response from Authors:

We thank the referee for pointing out this typo. To avoid confusion, we have replaced t with h to represent the substrate thickness.

Referee #1 Comment 11:

(3b) The math equations in lines 121 and 123 are incorrect in dimension. Please consider to correct them.

Response from Authors:

We thank the referees for pointing out this typo. We have corrected the math equations in the revised manuscript as below:

on page 5: The indirect coupling through the leaky wave is elucidated as $i\chi \propto \frac{\psi_{1(2)}^{nf} \cdot \psi_{2(1)}^p dV}{|C_{NF}| \cdot |C_{FF}|}$.

where normalization constants follow $|C_{NF}|^2 = \langle \psi_m^{nf} | \psi_m^{nf} \rangle_V$, and $|C_{FF}|^2 \delta(0) = \langle \psi_m^p | \psi_m^p \rangle_V$ ($m = 1, 2$).

on page 6: While the direct coupling is also proportional to the field overlap integral $\kappa \propto \frac{\int \psi_1^{nf} \cdot \psi_2^{nf} dV}{|C_{NF}|^2}$.

Since the coupling coefficients are normalized, we have also revised the corresponding coupled-mode equation (1) on page 5:

$$\omega \begin{bmatrix} \psi_1^{nf} \\ \psi_2^{nf} \end{bmatrix} = \begin{bmatrix} \omega_1 + i\gamma_0 & \omega_0(\kappa_{12} + i\chi_{12}) \\ \omega_0(\kappa_{21} + i\chi_{21}) & \omega_2 + i\gamma_0 \end{bmatrix} \begin{bmatrix} \psi_1^{nf} \\ \psi_2^{nf} \end{bmatrix}$$

where the direct coupling κ_{12} (κ_{21}) and indirect coupling χ_{12} (χ_{21}) are dimensionless parameters. Besides, we have modified all the expressions related to the above equations in the manuscript

and supplementary materials.

Referee #1 Comment 12:

(3c) In line 124, what "proper regions" should be? Please provide a quantitative estimation if possible.

Response from Authors:

We thank the referee for the valuable suggestion.

To provide a quantitative estimation of the proper region, we extract the indirect coupling strength from simulation results (in Fig. R1.10) as the coupling distance d changes. When the distance increases from 1 mm to 10 mm, the variation range of the indirect coupling strength $\omega_0\chi$ is only 0.003 GHz (from 0.024 GHz to 0.021 GHz). Such a small variation is negligible, so the indirect coupling strength can be approximately considered stable in the "proper region" of 1-10 mm (the blue region in Fig. R1.10).

As the coupling distance d further increases, the simulated indirect coupling strength decreases slightly. Such a reduction is attributed to the attenuation of the in-plane propagating component of leaky waves in the presence of the out-of-plane radiation leakage.

Fig.R1.10. The simulated indirect coupling strength χ as the coupling distance d between the two DSPRs evolves.

Regarding this estimation, we added this part on page 12 in the supplementary note II and included Fig. R1.5 as Fig. S2.

Referee #1 Comment 13:

(3d) It is unclear what advantages one can obtain with the space waves emphasized in this work.

Response from Authors:

We thank the reviewer for the critical comment.

To emphasize the advantages of space waves in our work, we have added two more new exotic APT phenomena based on our radiative plasmonic APT system, by exploring the DoFs of space

waves. (i) Switching polarizations of illuminating space waves, we realize **polarization-controlled APT phase transition**. (ii) By tuning incidence angles, we observed **multi-stage APT phase transition** in higher-order APT systems. Detailed information about the two phenomena is shown in response to comment 1a. Utilizing the advantages of space waves, we also showed the **remote sensing application**, i.e. far-field detection of environmental variations. The numerical results are included in the response to comment 1c.

Such results indicate that our radiative system enables us to explore the far-field DoFs of space waves for controlling or selectively exciting APT supermodes, which has not been explored in previous APT works. Attributed to the advantages of space waves, our scheme shows promise in constructing APT-empowered radiative devices and remote sensing applications.

Referee #1 Comment 14:

(3e) Reference issues. A few anti-PT experimental works have been overlooked by the authors. As they are highly relevant to the current work, we strongly encourage the authors to give them credits.

Response from Authors:

We thank the referees for reminding us of the missed references, which makes our introduction more comprehensive. We have added them in the reference part on pages 19-21 as below.

21. Jiang, Y., et al. Anti-Parity-Time Symmetric Optical Four-Wave Mixing in Cold Atoms. *Phys. Rev. Lett.* **123**, 193604 (2019)
22. He, Y., Wu, J., Hu, Y., Zhang, J.-X., & Zhu, S.-Y. Unidirectional reflectionless anti-parity-time-symmetric photonic lattices of thermal atoms. *Phys. Rev. A* **105**, 043712 (2022)
23. Ding, L., et al. Information retrieval and eigenstate coalescence in a non-Hermitian quantum system with anti-PT symmetry. *Phys. Rev. A* **105**, L010204 (2022)
25. Li, Q., et al. Experimental simulation of anti-parity-time symmetric Lorentz dynamics. *Optica* **6**, 000067 (2019)
28. Fan, H., Chen, J., Zhao, Z., Wen, J., & Huang, Y.-P. Antiparity-Time Symmetry in Passive Nanophotonics. *ACS Photonics* **7**, 3035-3041 (2020)
29. Bergman, A., et al. Observation of anti-parity-time-symmetry, phase transitions and exceptional points in an optical fibre. *Nat. Commun.* **12**, 486 (2021)
32. Wen, J., et al. Observation of information flow in the anti-PT-symmetric system with nuclear spins. *NPJ Quantum Inf.* **6**, 28 (2020)
34. Zhao, J., et al. Observation of Anti-PT-Symmetry Phase Transition in the Magnon-Cavity-Magnon Coupled System. *Phys. Rev. Appl.* **13**, 014053 (2020)

Referee #1 Comment 15:

In brief, as no new features or insights are available from the current work, I am not convinced that the manuscript meets the high standards of the journal. Moreover, the anti-PT features have been well understood in terms of phase transition and supermode dynamics.

Response from Authors:

We are grateful for the referee's critical comments. We have taken all of the suggestions into account and substantially advanced the work. We added **two more new exotic APT phenomena** by exploring the DoFs of space waves (polarizations and incident angles), i.e. **(i) polarization-controlled APT phase transition. (ii) multi-stage APT phase transition in high-order APT systems.** Both have not been reported in previous APT systems, and further strengthen the significance of our work. The **extension capability of our design to optical frequencies** has been verified with the numerical results. We also showed **promising applications in both near-field and far-field sensing environmental changes.** These new phenomena and extensions, we hope, have addressed the referees' concerns.

General comments from Referee #2:

Anti-parity-time (APT) symmetry is an emerging non-Hermitian mechanism, which has recently attracted intense interests from fundamental physics to technological applications. The authors of this manuscript propose a leaky-wave scheme for constructing radiative APT systems, and demonstrate it using designer-plasmonic structures. Conceptually, this work contributes to the photonic community a new concept i.e. radiative APT plasmonics, fundamentally different from conventional photonic APT systems for guided waves. Technologically, the leaky-wave scheme enables the realization of APT symmetry in the linear optics region, thus breaking the restriction of strong pumping light as previous photonic realizations based on nonlinear couplings. Moreover, the paper is well organized based on convincing results, covering theoretical analysis, numerical simulations, and experimental verifications. Overall, I think this work has the standards that one may expect for a publication in Nature Communications. I strongly recommend accepting this manuscript after addressing some minor concerns.

Response from Authors:

We thank the referees for recognizing both the **conceptual advancement** and **technological development** in our work. We also thank the referee for considering our work “well organized”, “convincing results”, and “recommend accepting”. Regarding the referee's suggestions, the point-by-point responses are provided as below.

Referee #2 Comment 1:

Minor concerns:

1. It is impressive that the 2-mm thicknesses of the samples are much smaller than the operational wavelengths. I am wondering whether that thickness achieves the limit.

Response from Authors:

We thank the referees for considering the subwavelength thickness *impressive*, which is promising in miniaturizing photonic APT devices.

Actually, we can further optimize the thickness of the substrate, but cannot always reduce it. If the thickness is further reduced, we need to consider its effect on the imaginary coupling strength. According to the theoretical expression of the imaginary coupling $i\chi \propto h \cdot$

$\int \frac{1}{|C_{NF}| \cdot |C_{FF}|} \psi_{1(2)}^{nf} \cdot \psi_{2(1)}^p dx dy$ (see the manuscript for the detailed analysis of the theoretical

expression), the imaginary coupling strength decreases as the thickness h decreases. The simulation result is also consistent with the theoretical analysis (in Fig. R2.1). Thus, when the substrate is too thin, the leaky-wave coupling tends to vanish. In this scenario, the APT system is approximately reduced to two independent high-Q resonators, and the system hardly interacts with space waves.

Overall, to ensure the efficient imaginary coupling by leaky waves, the thickness h of the dielectric substrate is elaborately designed. We can further optimize the thickness, but cannot always reduce it.

Fig. R2.1. The imaginary coupling strength χ as the thickness h of the substrate evolves.

Referee #2 Comment 2:

2. The corresponding frequency of the resonance mode in Fig. 2b should be given.

Response from Authors:

We thank the referee for reminding this issue. We have added the frequency of the resonance mode in the caption of Fig. 2b in the manuscript. The revision is on page 15 as below:

The E_z component of the simulated DSPR eigenmode (ψ^{n_f}) at 3.655 GHz on the XY (left) and XZ (right) planes.

Referee #2 Comment 3:

3. The field patterns of photonic APT systems are firstly captured in experiments, therefore showing the experimental setup would be helpful for reproducing.

Response from Authors:

We thank the referees for the constructive suggestion. According to the suggestion, we have added the schematics of both near-field and far-field experimental setups in Fig. R2.2.

Fig. R2.2. Schematics of near-field and far-field experimental setups. (a) Near-field experimental setup. (b) Far-field experimental setup

We also added the detailed description as below on page 14 in supplementary note V, and included Fig. R2.2 as Fig. S4:

The near-field and far-field experimental setups are shown in Fig. S4a-b. Experimental measurements are carried out in the microwave anechoic chamber using the vector network analyzer (VNA). In the near-field experiments, the sample is excited by the near-field source, and the probe is vertically suspended on a 2D moving stage controlled by the computer to capture the field patterns by detecting E_z components, avoiding interference from the incident waves. In the far-field experiments, we use the horn antenna as the far-field excitation source, and the probe is also vertically suspended on the 2D moving stage. It is worth noting that the odd mode of the DSPR is excited when the electric field of the incident wave is along the x direction, while plane waves with the electric field along the y direction excite the even mode.

Referee #2 Comment 4:

4. The references should be carefully selected, for example, [*Phys. Rev. Lett.* 123(19), 193604 (2019)] should be related to atomic APT systems and [*Nat. Commun.* 12(1), 486 (2021)] should be related to photonic APT systems. It is also suggested to cite [*Chinese Phys. B* 31(1), 014215 (2022)] when referring to enhanced Sagnac effect in APT systems.

Response from Authors:

We thank the referee for reminding us of these references, which makes our introduction more comprehensive. We have included these papers in the revised manuscript.

Referee #2 Comment 5:

5. At line 49, the full name of APT should be given.

Response from Authors:

We thank the referees for pointing out the typo. We have added the full name of APT in the

revised manuscript, i.e. anti-parity-time.

Referee #2 Comment 6:

6. Some of the references are missing page numbers, such as Ref. [9] at line 368, Ref. [15] at line 379, Ref. [20] at 388-389, and Ref. [25] at 397-398. Please check all the references carefully to avoid such typos.

Response from Authors:

We thank the referees for reminding us of these typos. We have checked all the references and completed the missing information in the revised manuscript.

REVIEWERS' COMMENTS

Reviewer #1 (Remarks to the Author):

In this revised manuscript, I am impressed by the extensive changes made by the authors, especially the new experimental results on demonstrating the advantages and new features of anti-parity-time symmetry based on their interesting systems. In comparison with their previous version, no doubt, this revised one looks much better in terms of quality, presentation, and significance.

I am happy with this new version of the manuscript, as one can indeed observe new findings and features that have not been disclosed in the past. Obviously, these properties originate from the platform proposed and demonstrated by the authors.

I am also very happy with the response from the authors in the sense that they not only carefully addressed all my comments and suggestions, but also added many and important information which are very helpful to clarify some confusion and circle out the importance of the claimed findings. Now, I agree with the authors' claim as the confusions are cleared out in their response as well as in the revision (both the main text and the supplementary materials). I notice that some of the materials written in the response have been added into the supplementary materials but some of them not. I encourage the authors to add more in the response into the supplementary so that a reader can gain a more complete understanding of the reported work.

In short, I am now convinced that the reported scheme does offer new features that have not been observed in the past. These new features are substantial and important and may lead to new technological innovations in wave control and manipulation. As the work falls into the scope of Nature Communications and also meets its high standards in terms of quality, significance and importance, I am now happy to recommend it for publication in the journal.

Reviewer #2 (Remarks to the Author):

I strongly recommend the publication of this manuscript in Nature communications.

Response Letter

We are grateful for the positive comments on our revised manuscript. In the text below, the referees' comments are quoted in *italics* and followed by our point-by-point response (in blue).

General comments from Referee #1:

In this revised manuscript, I am impressed by the extensive changes made by the authors, especially the new experimental results on demonstrating the advantages and new features of anti-parity-time symmetry based on their interesting systems. In comparison with their previous version, no doubt, this revised one looks much better in terms of quality, presentation, and significance.

I am happy with this new version of the manuscript, as one can indeed observe new findings and features that have not been disclosed in the past. Obviously, these properties originate from the platform proposed and demonstrated by the authors.

I am also very happy with the response from the authors in the sense that they not only carefully addressed all my comments and suggestions, but also added many and important information which are very helpful to clarify some confusion and circle out the importance of the claimed findings. Now, I agree with the authors' claim as the confusions are cleared out in their response as well as in the revision (both the main text and the supplementary materials). I notice that some of the materials written in the response have been added into the supplementary materials but some of them not. I encourage the authors to add more in the response into the supplementary so that a reader can gain a more complete understanding of the reported work.

In short, I am now convinced that the reported scheme does offer new features that have not been observed in the past. These new features are substantial and important and may lead to new technological innovations in wave control and manipulation. As the work falls into the scope of Nature Communications and also meets its high standards in terms of quality, significance and importance, I am now happy to recommend it for publication in the journal.

Response from Authors:

We are happy that we have addressed all the comments and suggestions. We thank the referee for recognizing the significance of our work and recommending it for publication.

According to the referee's suggestion, we have added other important information in the first-round response to the supplementary information (on page 25).

Referee #2 Comment 1:

I strongly recommend the publication of this manuscript in Nature communications.

Response from Authors:

We thank the referee for "strongly recommending the publication of this manuscript".